# A Pilot Study to Examine the Impact of Beta-Alanine Supplementation on Anaerobic Exercise Performance in Collegiate Rugby Athletes

**DOI:** 10.3390/sports7110231

**Published:** 2019-11-07

**Authors:** Charles R. Smith, Patrick S. Harty, Richard A. Stecker, Chad M. Kerksick

**Affiliations:** 1Arnold School of Public Health, University of South Carolina—Columbia, Columbia, SC 29208, USA; cs36@email.sc.edu; 2Energy Balance and Body Composition Laboratory, Department of Kinesiology & Sport Management, Texas Tech University, Lubbock, TX 79409, USA; Patrick.Harty@ttu.edu; 3Exercise and Performance Nutrition Laboratory, School of Health Sciences, Lindenwood University, St. Charles, MO 63301, USA; rstecker@lindenwood.edu

**Keywords:** ergogenic aid, high intensity, fatigue, buffering, lactate

## Abstract

Beta-alanine (BA) is a precursor to carnosine which functions as a buffer assisting in the maintenance of intracellular pH during high-intensity efforts. Rugby is a sport characterized by multiple intermittent periods of maximal or near maximal efforts with short periods of rest/active recovery. The purpose of this pilot study was to evaluate the impact of six weeks of beta-alanine supplementation on anaerobic performance measures in collegiate rugby players. Twenty-one male, collegiate rugby players were recruited, while fifteen completed post-testing (Mean ± SD; Age: 21.0 ± 1.8 years, Height: 179 ± 6.3 cm, Body Mass: 91.8 ± 13.3 kg, % Body Fat: 21.3 ± 4.4). Supplementation was randomized in a double-blind, placebo-controlled manner between 6.4 g/d of beta-alanine and 6.4 g/d of maltodextrin placebo. Body composition, upper and lower-body maximal strength and muscular endurance, intermittent sprint performance, and post-exercise lactate, heart rate, and rating of perceived exertion were assessed before and after supplementation. Data were analyzed using a 2 × 2 (group × time) mixed factorial analysis of variance (ANOVA) with repeated measures on time. No significant interaction effects were noted for body mass, fat mass, fat-free mass, and percent bodyfat (*p* > 0.05). No performance effects resulting from beta-alanine supplementation were detected. Results from this initial pilot investigation suggest that BA exerts little to no impact on body composition parameters, muscular strength, muscular endurance, or intermittent sprinting performance. With the limited research exploring the impact of BA in this sporting context, these initial findings offer little support for BA use, but more research is needed to fully understand the potential impact of BA on various aspects of resistance exercise performance.

## 1. Introduction

Rugby is a sport characterized by periods of maximal- or near-maximal exertion interspersed with periods of low- to moderate-intensity exercise or recovery [1,2]. Blood lactate concentrations of 5.9–8.4 mM have been documented during simulated rugby play reflecting high anaerobic demands in conjunction with remarkable acidosis, which may function as a limiting factor to rugby performance [3]. Increased lactate concentrations are correlated with the accumulation of hydrogen ions (H^+^) [4], therefore the maintenance of intracellular pH may help in the delay of fatigue in addition to overall physical performance during rugby. Carnosine (beta-alanyl-L-histidine) is an intramuscular dipeptide made up of L-histidine and beta-alanine [5]. Carnosine acts as an important intramuscular buffer and exists in high concentrations in muscle tissue [6,7,8]. Studies have indicated that increases in intramuscular carnosine enhances the body’s ability to sustain high intensity exercise performance, an outcome that is purported to be due to increased buffering of intramuscular H^+^ [6]. This process is thought to increase muscle contractility while simultaneously maintaining proton gradients and attenuating intramuscular acidosis. Notably, due to the presence of carnosinase in human plasma, oral ingestion of carnosine fails to increase intramuscular carnosine content, however, intramuscular carnosine content can be increased to appreciable degrees after supplementation with beta-alanine [8,9,10,11].

Beta-alanine is recognized as the rate-limiting factor of carnosine synthesis due to its existence in remarkably low concentrations in skeletal muscle, while the other component of carnosine (L-histidine) is extremely abundant in skeletal muscle [5]. Oral supplementation with beta-alanine at a dosage of 4–6 g/day over the course of 4–10 weeks has been shown to increase intramuscular carnosine content by 40–80% [6,7,8,11,12]. Consequently, beta-alanine supplementation has become an increasingly popular strategy for athletes performing high-intensity, maximal activity and appears to be most effective in open-ended activities lasting 60–240 s (s) in duration at very high intensities where acidosis accumulates inside the cell [8,13]. Indeed, 4–10 weeks of beta-alanine supplementation has been shown to increase total work completed at 110% maximal power and 30 s sprint capacity after a two-hour time trial [12]. Furthermore, increases are also seen in total work performed with repeated 30 s bouts of maximal upper body ergometry following beta-alanine supplementation [14]. While multiple studies have indicated beta-alanine supplementation may improve different forms of high-intensity exercise, little data exists with resistance-based exercise and high-intensity anaerobic exercise. In this respect, research has indicated that beta-alanine supplementation may improve force production during maximal voluntary isometric contractions [15,16,17] or isokinetic exercise performance [15,18]. While these increases are meaningful, these modes of exercise performance may not directly translate into improvements in sport performance. In this respect, Hoffman and colleagues [19] showed that beta-alanine supplementation (6.4 g/day over 30 days) increased total training volume and reduced rating of perceived exertion (RPE) and fatigue in collegiate football athletes compared to placebo, but its effect on football-specific performance measures was not evaluated. As it stands, more research is needed to examine the ability of beta-alanine to influence anaerobic or resistance exercise performance in trained athletes. Therefore, the purpose of this study was to investigate the effects of six weeks of beta-alanine supplementation at a dosage of 6.4 g/day on anaerobic and resistance exercise performance in a group of trained collegiate athletes.

## 2. Materials and Methods

### 2.1. Experimental Approach to the Problem

This trial was conducted as a pilot approach to a potentially larger trial. All testing occurred over two consecutive days (Figure 1). On the first day, participants were asked to refrain from exercise for 24 h and arrive after observing at least an eight-hour fast from all calorie-containing foods and drinks. Upon arrival, participants were provided the informed consent documentation and an oral review of the study goals, schedule, and testing procedures. Following signed consent, body mass was determined and body composition using dual-energy x-ray absorptiometry (DEXA) were evaluated before completing a standardized warm-up. After completion of the warm-up, participants completed a repeated sprint test composed of six sets of 30 s all-out shuttles with 30 s rest intervals in between sets [2]. Bench press one-repetition maximum (1 RM) was initially determined followed by determination of back squat 1 RM with ten minutes rest provided between each exercise [20]. The second day of testing was completed the next day at a similar time. On this day and after completion of a standardized warm-up consisting of whole-body movement and flexibility activities, upper and lower body strength endurance were assessed using a five set to fatigue protocol at 70% 1 RM for bench press and back squat [21,22]. Repetitions completed during each set were recorded. Tests were completed in a randomized manner with two minutes rest between sets and ten minutes rest between tests. After completion of the second day of testing, participants were randomized in a double-blind, placebo-controlled manner to ingest either 6.4 g/day of placebo (maltodextrin) or beta-alanine capsules. Each participant was required to ingest four daily doses of 1.6 g per dose over the course of a six-week supplementation protocol. Participants were required to record all food and fluid intake over a four-day period at the beginning of the study and were instructed to not change their dietary or hydration patterns throughout the remainder of the study. Dietary intake was likewise assessed during the four-day period prior to post-testing. All participants then completed weekly strength and conditioning sessions with a certified strength and conditioning specialist along with weekly team practices and drills. All participants returned to the laboratory after six weeks of supplementation to complete an identical testing battery as what was completed at baseline.

### 2.2. Subjects

Twenty-one male subjects were recruited from the active roster of the university men’s rugby team, which competes at the Division IA level. Of the initial participants recruited, 15 were included in the final analysis (Mean ± SD; Age: 21.0 ± 1.8 years, Height: 179 ± 6.3 cm, Body Mass: 91.8 ± 13.3 kg, % Body Fat: 21.3% ± 4.4% fat). All participants were actively engaged in weekly team-based strength and conditioning sessions as well as weekly team practices focusing upon strategy and conditioning. As determined by the review of health history and medical records, participants were required to be healthy and free of disease. All participants abstained from taking any additional forms of nutritional supplementation (with the exception of protein and a multi-vitamin/mineral) for four weeks prior to beginning the study and for the entire duration of the study. Participants were excluded if they were currently diagnosed with or being treated in any capacity for any cardiovascular, renal, pulmonary, orthopedic, immunological, psychological, or musculoskeletal disorder. Any individuals who were currently taking any prescription or over-the-counter medications known to impact metabolic responses to exercise were excluded as were any participant who failed to stop taking any dietary supplements leading up to and throughout the study protocol. This study was approved by the Lindenwood University Institutional Review Board (IRB; Protocol 939436-1; Approved 22 August 2016), and informed consent was provided by all participants prior to their participation.

### 2.3. Procedures

#### 2.3.1. Anthropometry

Body mass was assessed using a TANITA electronic scale (BWB 627-A Class III, TANITA, Arlington Heights, IL, USA). Height was assessed using a wall-mounted stadiometer (HR-200, TANITA, Arlington Heights, IL, USA) with shoes off, feet together, and against the wall. These measures were collected upon arrival and prior to DEXA scans [23] and were used for both demographic purposes and as information required by the DEXA software (Version 4.5.3, HOLOGIC, Bedford, MA, USA) for accurate analysis of body composition.

#### 2.3.2. Dual-Energy X-Ray Absorptiometry (DEXA)

To standardize pre-testing conditions prior to completing the DEXA, participants were instructed to arrive after observing an eight-hour fast of energy-containing food and fluid and avoiding exercise for at least 24 h [23]. Hydration status was not overtly controlled or standardized as participants were instructed to not restrict their water intake prior to having a DEXA performed. Calibration procedures were completed each day before testing and all DEXA scans were performed using a Hologic QDR Discovery A (HOLOGIC, Bedford, MA, USA) and analyzed using its accompanying software (Hologic APEX Software, Version 4.5.3, HOLOGIC, Bedford, MA, USA) to determine whole-body levels of bone, fat, and fat-free masses, along with body fat percentages. The TBAR 1209 (Pre-NHANES) calibration method was applied to all assessments. All DEXAs were performed and analyzed according to the manufacturer guidelines by the same two laboratory investigators for the entire study. Test-retest reliability of performing this test on 20 healthy, college-aged individuals on our equipment yielded intra-class correlation coefficients of ≥0.998.

#### 2.3.3. 1 RM Determination

After body composition assessment, 1 RM testing was completed first using the bench press exercise and was followed with the back-squat exercise. Testing procedures were based upon National Strength and Conditioning Association (NSCA) guidelines as outlined by Haff et al. [20]. Briefly, the participants first completed warm-up sets of ten, five, and two repetitions at self-selected, ascending loads to build towards a 1 RM load. Participants then completed up to five one-repetition attempts to determine their 1 RM. Three minutes of rest was provided between each set and ten minutes rest was provided between bench press and back squat testing.

#### 2.3.4. Upper Body and Lower Body Strength Endurance Testing

The second day of testing was completed at a similar time and consisted of modified strength endurance according to the methods of Trepanowski et al. [22] and Ferguson et al. [21]. These tests entailed completing five sets of bench press and back squat repetitions to fatigue at a load equivalent to 70% of their 1 RM. Fatigue was defined as either a two second pause at any point in the set, either between repetitions or at any point in the motion, or volitional/muscular fatigue whereby the participant elected to end the set or the repetition could not be completed. Two minutes of rest were allotted between sets to allow for minor recovery before the next set. Ten minutes of rest were allotted between each test to allow for muscular recovery. Upper and lower body strength endurance testing were completed in randomized order between participants to account for any fatigue that may cross-over between exercises. Each participant completed the exercises during post-testing in the same order as they did pre-testing to allow for consistency between pre- and post-testing. Repetitions completed were recorded after each set, and total repetitions completed were assessed after test completion.

#### 2.3.5. Intermittent Running Test

The intermittent running test required subjects to perform a series of six 30 s sprints whereby the athlete covered as much distance as possible during each of the six sprints. For each sprint, the athlete traversed incrementally increasing distances: 5, 10, 15, 20, and 25 yards. At each distance the athlete returned to the starting position before immediately beginning the next distance. The athlete completed this pattern in a continual fashion until 30 s had elapsed. If the athlete was able to traverse all distances before time ran out, they started the pattern over. Upon completion of each 30 s sprint, the athletes rested for 30 s. A total of six sprints were completed with 30 s rest between each bout. The distance covered was closely observed by study investigators to estimate the distance covered when time expired. Total distance for each sprint along with changes in blood lactate were calculated for the entire test.

#### 2.3.6. Blood Lactate Determination

Capillary blood samples were collected after sterilization with rubbing alcohol and a sterilized lancet. Each blood sample (~250 µL) was collected and placed onto a lactate analysis strip and analyzed for lactate concentration. The lactate analyzer (Lactate Plus™, www.lactate.com) was calibrated according to manufacturer guidelines using high- and low-lactate control solutions. The Lactate Plus analyzer has been used in a number of previous exercise physiology applications and has been shown to provide reliable lactate results [24]. Lactate concentrations were determined prior to having the DEXA scan completed (Rest) and after 2nd, 4th, and 6th sets of completing the intermittent running test.

#### 2.3.7. Heart Rate and Rating of Perceived Exertion (HR and RPE)

Heart rate (HR) and rating of perceived exertion (RPE) were collected following every set of the intermittent running test. Heart rate was measured using heart rate monitors (Polar TI34-Coded, Lake Success, NY, USA), whereby the subject wore a watch on one wrist which wirelessly connected to a moistened monitoring strap which was placed securely and snugly around the torso at approximately the level of the xiphoid process. RPE representing the difficulty of the work completed during each set was quantified by the athlete using a modified Borg Scale where 1 = effortless and 10 = absolute maximum effort [25].

### 2.4. Supplementation

Both beta-alanine and placebo supplements were provided by CarnoSyn (National Alternatives International, San Marcos, CA, USA). Following completion of the second day of testing, participants were assigned their supplement in a randomized double-blinded manner. The subjects ingested either 6.4 g/d of a time-release formulation of beta-alanine (*n =* 8) or a maltodextrin placebo (*n* = 7) in four equal doses of 1.6 g. Each 1.6-g dose was consumed with food and was separated by at least two hours to limit any paresthesia from beta-alanine administration [8]. This dose was chosen due to the well-documented effects of 6 g/day beta-alanine administered for four weeks [8]. The participants were instructed to follow this supplementation protocol and their normal training and dietary habits/routines for the next six weeks with their last dose being taken 90 min prior to day two of post-testing. If a dose was skipped or missed, participants were instructed to make the dose up within the same day while leaving at least two hours between subsequent doses. A four-day diet log was collected prior to pre- and post-testing to ensure consistent energy intake prior to testing.

### 2.5. Training Program

All participants completed regular weight training and conditioning sessions with a strength and conditioning coach three days per week for at least one hour per session. These practices involved total body resistance training and conditioning work specific to the sport of rugby. The participants also had practice with their sport coach four days per week for at least two hours per session. These practices included sport-specific drills, game-simulated activities, and more conditioning work. Participants were provided with two rest/recovery days per week which included review of game film in preparation for the season. This routine was continuous over the six-week supplementation period.

### 2.6. Statistical Analyses

All data are presented as means ± SD. Descriptive statistics were calculated using standard statistical methods. Shapiro–Wilk tests were employed to assess normality of all outcome variables. In cases where data were determined to be non-normal, log transformations, square root transformations, and cube transformations were computed, and the data were re-assessed for normality following each transformation. In cases where transformations improved model assumptions, transformed data was used. Then, 2 × 2 (group × time) mixed factorial ANOVAs with repeated measures on time were used to identify the presence of any main and interaction effects. Ninety-five percent confidence intervals of the estimated marginal means were calculated for all outcome measures. Effect size (Cohen’s *d*) was determined for all outcome measures using the following formula: *d* =  (Change_Beta-Alanine_ − Change_Control_)/SD_pooled_ [26]. Effect sizes were classified as small (*d* < 0.2), medium (0.2 < *d* < 0.5), or large (0.5 < *d* < 0.8) [27]. The threshold of statistical significance was set at α = 0.05. Microsoft Excel was used to organize all data and the Statistical Package for the Social Sciences (Version 23, IBM Corporation, Armonk, NY, USA) was used to complete all statistical analysis.

## 3. Results

Of the twenty-one participants who completed pre-testing, fifteen completed post-testing. Two were lost due to injuries, three were lost due to noncompliance, and one withdrew from the study. The three individuals who were lost due to noncompliance were removed from analysis in the final dataset due to their failure to comply with maximal effort during post-testing despite detailed instruction and encouragement by the team members. These individuals were interviewed by team members and disclosed an unwillingness to give maximal effort due to their fear of injuries, fatigue, and soreness for an upcoming match against a rival team. In cases where the included participants were unable to complete post-testing, the baseline values were carried forward and imputed as post-testing values.

### 3.1. Dietary Analysis

Dietary recall data was available from 13 of the 15 participants (intervention group: *n* = 7; control group: *n* = 6.) No significant main effects or interaction effects were identified for total kilocalories consumed (Pre: 2450 ± 363; Post: 2210 ± 806 kcal), relative kilocalories consumed (Pre: 26.7 ± 5.2; Post: 24.6 ± 10.2 kcal/kg bodyweight), total carbohydrates consumed (Pre: 256 ± 85; Post: 228 ± 112 g), relative carbohydrates consumed (Pre: 2.8 ± 1.1; Post: 2.6 ± 1.5 g/kg bodyweight), total protein consumed (Pre: 124.8 ± 31.6; Post: 112 ± 45.4 g), relative protein consumed (Pre: 1.4 ± 0.4; Post: 1.3 ± 0.6 g/kg bodyweight), total fat consumed (Pre: 97.2 ± 17.6; Post: 80.5 ± 30.6 g), and relative fat consumed (Pre: 1.1 ± 0.2; Post: 0.9 ± 0.4 g/kg bodyweight).

### 3.2. Body Composition

No significant interaction effects were identified for body mass, percent body fat, fat mass, or fat-free mass (Table 1). However, a main effect for time was identified for percent body fat (*p* < 0.001) and fat mass (*p* = 0.001), suggesting that both groups decreased in fat mass and percent body fat across the intervention.

### 3.3. Muscular Strength/Endurance

No significant main or interaction effects were identified for bench press 1 RM, back squat 1 RM, bench press total repetitions completed, back squat total repetitions completed, and total volume load completed during bench press or back squat (Table 2).

### 3.4. Intermittent Sprint Performance and Recovery

There were no significant interaction effects observed for total distance covered during the intermittent running test, recovery heart rate following the test, recovery blood lactate following the test, or recovery RPE (Table 3). However, a significant main effect for time (*p* = 0.029) was observed for recovery blood lactate, suggesting that both groups were able to more effectively buffer lactate during post-testing. In addition, a main effect for group (*p* = 0.001) was noted for RPE.

## 4. Discussion

The primary findings of this pilot study provide more evidence regarding the potential impact of beta-alanine on anaerobic performance. Overall, beta-alanine’s ability to exert ergogenic outcomes on maximal strength and repetitions to fatigue appears limited as changes in strength, endurance, and power were lacking. These results fall in line with previous observations, as several studies have generally failed to report ergogenic outcomes in these parameters following beta-alanine supplementation [8]. Further, this pilot investigation suggested a limited potential for beta-alanine to improve muscular endurance. These outcomes are paralleled by previous studies that indicate beta-alanine can help with increasing training volume over the course of a training cycle [19], however, any improvement in volume has yet to consistently be translated into improvements in performance or body composition [8,16,18,28]. In addition, changes in RPE were similar between supplementation groups, suggesting minimal effect of beta-alanine supplementation on subjective measures of exercise intensity and difficulty.

Beta-alanine supplementation exerted no impact on body mass or body composition changes in comparison to those seen with placebo. This outcome mirrors the effects shown by Outlaw et al. [16] and Smith et al. [29]. Some key points to consider before interpreting these outcomes, however, are that all studies were on the short end (3–6 weeks) of the duration one could expect see measurable changes in body composition parameters. Additionally, while all three studies invoked well-constructed exercise programs, neither the aforementioned studies nor the present investigation employed a training program where maximizing body composition changes was a primary consideration. It is possible these results may have been different had we chosen to more closely examine strength and power changes in a dedicated strength and conditioning program during the team’s off-season. While a valuable research question to explore, this was not a primary consideration for the current study.

Initial research in laboratory settings has demonstrated the ability of beta-alanine to increase work capacity in fatiguing or exhausting tasks [13], while studies that employed more sport specific or practical measures are lacking. In this pilot study, we employed a multi-set repetitions to fatigue approach using two popular resistance training exercises to explore the ergogenic impact of beta-alanine. Under these delimitations, results from the present study indicate no difference between the beta-alanine and placebo groups in maximal strength or muscular endurance as evaluated by the number of repetitions completed after each set. However, a trend for a group x time interaction was identified for bench press 1 RM (*p* = 0.067, *d* = 0.39, small), suggesting that beta-alanine supplementation may have merit to improve upper body force production. However, previous work has demonstrated increases in total work completed, time to exhaustion, and increased training volume [13,14,17,19,28]. Briefly, a number of factors may explain these divergent outcomes. For example, an unbroken stimulus of at least 60 s may be required for beta-alanine to yield an ergogenic outcome [13,17,30], a duration of high-intensity work not commonly seen by many team-sport, strength-and-power athletes. Moreover, it is likely that beta-alanine may operate most effectively in open-ended tasks that span at least 60 continuous seconds rather than completion of multiple, shorter maximal bouts that when summed are greater than 60 s [8,13]. Also, the total duration of both strength endurance protocols was longer than the previously indicated upper ergogenic threshold of 240 s where benefits of the substance become less pronounced. By design, our pilot approach did not use any single continuous stimulus lasting within the desired 60–240 s window where ergogenic outcomes are commonly observed for beta-alanine. For this primary reason, we feel, the potential ergogenic effects of beta-alanine were not observed [8,13]. In this regard, another contributing factor could have been the two-minute rest period that was provided between each set, which may have allowed an appropriate amount of recovery between each set that negated the benefits of beta-alanine. Certainly, while shorter or longer rest periods, different exercise choices, or set/rep schemes may have led to different outcomes, it remains that our preliminary findings indicate beta-alanine offered minimal ergogenic potential within the practical setting described in this study. Importantly, these were a priori decisions made in this pilot investigation to use traditional resistance exercise programming to assess the potential for beta-alanine to operate in an ergogenic fashion. This area is largely unexplored yet, beta-alanine continues to be a common ingredient in pre-workout formulations and employed by athletes who train and compete in this manner.

In contrast to the results of previous investigations [8], no effect of beta-alanine was found on anaerobic performance, blood lactate responses, heart rate responses, or RPE during recovery. However, it is important to note that a beta-alanine exerted large but non-significant effects on recovery blood lactate (*d* = −0.94) and recovery RPE (*d* = −1.23), suggesting that the supplement may have some degree of practical benefit for acute recovery following anaerobic exercise. As with the strength tests employed in the present investigation, the test duration of the intermittent running test of 360 s falls outside the 240 s “ceiling” of beta-alanine’s maximal effectiveness. Also, each sprint only lasted 30 s which falls far below the “minimal” time requirement for beta-alanine to be effective. Therefore, even though the total stimulus across all six sets of sprints totals 3 min or 180 s, which is in the middle of beta-alanine’s window of effectiveness, it would appear a continuous stimulus lasting 60–240 s is required for beta-alanine to be most effective. Thus, while our chosen total exercise stimulus may have exceeded the previously proposed ceiling for the ergogenic benefit of beta-alanine, we failed to observe performance improvements after any of the running bouts within the intermittent running test. In addition, a pacing/learning effect may have influenced the results of this test, as the athletes may have developed strategies to help attenuate fatigue and acidosis. In light of these findings, more work is needed to expand upon our pilot investigation with better powered investigations to more suitably examine the impact of beta-alanine under these conditions.

The largest limitation to this study was the non-compliance and attrition rate, which reduced our sample size and statistical power, potentially resulting in some of the null findings which were observed. While this study was developed with a pilot approach, the attrition was larger than anticipated. The primary reasons for this attrition involved logistics and the time in the academic calendar when testing occurred as pre-testing occurred during the first week of school and post-testing adjacent to a competition that same weekend as well as mid-term testing in classes. As expected of this sport, we also had anticipated attrition due to injuries that occurred as part of competition. It is important to highlight that in spite of these challenges, our final sample size is similar to that used in previous studies [15,16,17,18] and with all other approaches taking to limit internal variability, we failed to see ergogenic outcomes for beta-alanine. Another limitation is that athletes were not prescribed a specific exercise protocol by the research team and instead were allowed to follow their normal personal/team training routine. While all team members were expected to participate in all drills and activities, some athletes may have had a larger training stimulus or volume over the course of the study, but because this was spread across all athletes involved, we feel this limitation would exert minimal impact on our outcomes.

The largest strength of the study lies within the overall study approach and methodologies employed. This study evaluated beta-alanine supplementation in an athletic population using sport-specific measures of performance as opposed to laboratory-based approaches. This subject area regarding this supplement is grossly under-represented in current literature [8,9,13,31]. In addition, this study is one of the few to utilize dynamic, free-weight resistance exercise tests [16] rather than isometric or isokinetic dynamometry measures to evaluate performance, measures which are more specific to the weight room/coaching setting [15,17,18]. We do feel that the loss of sensitivity was appropriately countered by a higher level of practicality and ecological validity for a competing strength and power athlete. Future research should continue to evaluate the supplement’s efficacy in these environments to further elucidate beta-alanine’s efficacy as an ergogenic aid in sport-specific settings. In addition, more information is needed concerning the impact of timed administration of beta-alanine on performance outcomes and training adaptations in team sport athletes.

In conclusion, results from this study and other published research continue to indicate that beta-alanine offers the potential to improve recovery after bouts of high-intensity exercise. However, our results suggest that beta-alanine has minimal impact on body composition, maximal strength, and strength-endurance. As one of the first steps to more rigorously examine the impact of beta-alanine on resistance/anaerobic style exercise performance, further research is required utilizing sport-specific measures to better ascertain if beta-alanine can offer meaningful benefits as an ergogenic aid. Since many anaerobic athletes complete resistance training to improve strength and power, but also may complete other forms of training (sprinting, intervals, or other forms of anaerobic conditioning) and compete in a manner where beta-alanine may afford some benefit beyond what was measured in the present study. In this respect, further research is required utilizing sport-specific measures to assess other performance variables, as the areas where beta-alanine may afford benefit to an anaerobic athlete may be more specific and nuanced.

## Figures and Tables

**Figure 1 sports-07-00231-f001:**
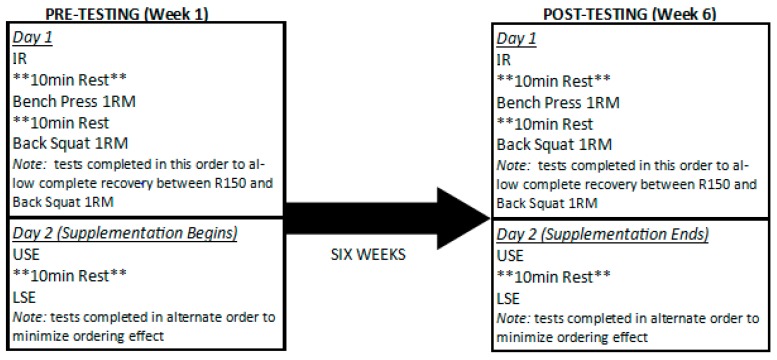
Overview of study design. IR: intermittent running test; 1 RM: one-repetition maximum test; USE: upper body strength endurance test; LSE: lower body strength endurance test.

**Table 1 sports-07-00231-t001:** Body composition changes following beta-alanine supplementation.

Outcome	Group	Baseline (95% CI)	Post-Testing (95% CI)	*d*	Group (*p*)	Time (*p*)	G × T (*p*)
Body Mass (kg)	**β-alanine**	89.3 ± 14.1 (79.0–99.6)	88.0 ± 11.7 (79.4–96.7)	0.02	0.416	0.064	0.895
	**Placebo**	94.8 ± 12.7 (83.8–105.8)	93.3 ± 11.0 (84.1–102.6)				
Percent Bodyfat	**β-alanine**	20.9 ± 4.4 (17.4–24.4)	19.7 ± 4.5 (16.4–23.1)	0.07	0.789	<0.001	0.532
	**Placebo**	21.7 ± 4.8 (17.9–25.4)	20.2 ± 4.2 (16.6–23.8)				
Fat mass (kg)	**β-alanine**	18.3 ± 6.7 (13.0–23.6)	16.9 ± 6.0 (12.5–21.4)	0.06	0.574	0.001	0.869
	**Placebo**	20.2 ± 7.2 (14.5–25.9)	18.4 ± 5.8 (13.6–23.2)				
Fat-free mass (kg)	**β-alanine**	66.7 ± 8.0 (61.0–72.3)	67.4 ± 6.0 (63.1–71.8)	0.15	0.121	0.825	0.540
	**Placebo**	72.7 ± 6.6 (66.7–78.8)	72.4 ± 5.3 (67.7–77.0)				

All data presented as Mean ± SD. Kg: kilogram; 95% CI: 95% confidence interval; d = Cohen’s d; *p*: *p*-value; Group: main effect for group; Time: main effect for time; G × T: group by time interaction.

**Table 2 sports-07-00231-t002:** Muscular strength and endurance following 6 weeks of beta-alanine supplementation.

Outcome	Group	Baseline (95% CI)	Post-Testing (95% CI)	*d*	Group (*p*)	Time (*p*)	G × T (*p*)
Bench Press 1 RM (kg)	**β-alanine**	104.0 ± 20.6 (87.2–120.9)	108.7 ± 14.2 (94.7–122.8)	0.39	0.071	0.722	0.067
	**Placebo**	128.3 ± 23.8 (110.3–146.3)	125.1 ± 22.3 (110.1–140.1)				
Back Squat 1 RM (kg)	**β-alanine**	137.8 ± 30.3 (113.9–161.7)	140.5 ± 27.5 (120.9–160.1)	0.12	0.332	0.726	0.517
	**Placebo**	154.2 ± 32.4 (128.7–179.8)	153.4 ± 23.3 (132.5–174.3)				
Total Bench Press Repetitions Completed	**β-alanine**	50.9 ± 10.2 (45.0–56.7)	46.9 ± 10.2 (40.7–53.0)	−0.93	0.001	0.872	0.118
	**Placebo**	32.7 ± 2.3 (26.5–38.9)	36.0 ± 4.4 (29.4–42.6)				
Total Back Squat Repetitions Completed	**β-alanine**	43.5 ± 14.5 (34.6–52.4)	47.9 ± 15.8 (38.4–57.3)	0.33	0.589	0.150	0.231
	**Placebo**	42.1 ± 6.8 (32.7–51.6)	42.6 ± 6.2 (32.5–52.7)				
Bench Press Total Volume Load (kg)	**β-alanine**	3654.0 ± 717.1 (3146–4161)	3485.9 ± 658.6 (3029–3943)	−0.59	0.103	0.867	0.127
	**Placebo**	2929.2 ± 597.0 (2387–3472)	3136.6 ± 518.3 (2648–3625)				
Back Squat Total Volume Load (kg)	**β-alanine**	4186.7 ± 1494.9 (3245–5129)	4626.9 ± 1526.6 (3649–5605)	0.35	0.881	0.193	0.202
	**Placebo**	4500.6 ± 830.2 (3493–5508)	4505.5 ± 911.9 (3460.2–5551)				

All data presented as Mean ± SD. Kg: kilogram; 95% CI: 95% confidence interval; *d* = Cohen’s d; *p*: *p*-value; Group: main effect for group; Time: main effect for time; G × T: group by time interaction.

**Table 3 sports-07-00231-t003:** Intermittent sprint performance and recovery.

Outcome	Group	Baseline (95% CI)	Post-Testing (95%CI)	*d*	Group (*p*)	Time (*p*)	G × T (*p*)
Total Distance Covered (m)	**β-alanine**	648.7 ± 27.5 (620.9–676.4)	677.2 ± 40.1 (645.2–709.3)	0.01	0.114	0.004	0.976
	**Placebo**	617.2 ± 44.5 (587.6–646.9)	645.3 ± 44.0 (611.0–679.6)				
Recovery Heart Rate (BPM)	**β-alanine**	183.3 ± 9.0 (175.8–190.8)	180.1 ± 11.6 (172.3–188.0)	0.29	0.263	0.120	0.623
	**Placebo**	189.6 ± 9.2 (182.1–197.0)	183.7 ± 6.9 (175.8–191.6)				
Recovery Blood Lactate (mmol)	**β-alanine**	13.4 ± 2.0 (11.8–15.1)	10.3 ± 3.5 (8.0–12.6)	−0.94	0.655	0.029	0.126
	**Placebo**	12.7 ± 2.4 (10.9–14.5)	12.1 ± 2.4 (9.6–14.6)				
Recovery RPE (0–10)	**β-alanine**	8.8 ± 0.7 (8.2–9.3)	8.0 ± 1.1 (7.3–8.7)	−1.228	0.001	0.366	0.191
	**Placebo**	9.4 ± 0.8 (8.8–10.0)	9.6 ± 0.5 (8.9–10.3)				

All data presented as Mean ± SD. 95% CI: 95% confidence interval; *d* = Cohen’s d; *p*: *p*-value; Group: main effect for group; Time: main effect for time; G × T: group by time interaction.

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
