# Peer review of "A Pilot Study to Examine the Impact of Beta-Alanine Supplementation on Anaerobic Exercise Performance in Collegiate Rugby Athletes"

_sports, 2019, doi:10.3390/sports7110231_

Round 1

Reviewer 1 Report

The purpose of this study was to evaluate the impact of six weeks of beta-alanine 22 supplementation on anaerobic performance measures in collegiate rugby players. Twenty-one male, 23 collegiate rugby players were recruited, while fifteen completed post-testing.

Generally, the importance of this investigation was well recognized and well deserved.

The abstract covers strictly the main statements of the publication, Introduction is short but concise, Methods are adequate for the purposes.

However, the narration of introduction dealing with the specific mechanism about beta-alanine supplementation and football-specific performance. and it also needs the field tests (football). More subjects are needed in this experiments to evaluate this purpose.

It needs to be described about specific rationale (dose od beta-alanine )and references in the section of methods.

It also needs of diverse dose groups.

Further study ie required utilizing diverse dose (diverse timing) of beta-alanine supplementation on foot ball performance (field test). 

Author Response

Author Responses

Note: All changes to the manuscript are highlighted in yellow.

Reviewer 1

The purpose of this study was to evaluate the impact of six weeks of beta-alanine 22 supplementation on anaerobic performance measures in collegiate rugby players. Twenty-one male, 23 collegiate rugby players were recruited, while fifteen completed post-testing. Generally, the importance of this investigation was well recognized and well deserved. The abstract covers strictly the main statements of the publication, Introduction is short but concise, Methods are adequate for the purposes.

However, the narration of introduction dealing with the specific mechanism about beta-alanine supplementation and football-specific performance. and it also needs the field tests (football). More subjects are needed in this experiments to evaluate this purpose.

Author Response: Thank you for your appraisal of our manuscript. Our sport was not football and involved competitive, collegiate rugby athletes.  The intermittent running test included as part of our protocol was a conditioning test performance by the team throughout their training calendar.  We recognize the limitations of our sample size and have highlighted that throughout.  Our goal was to conduct an initial pilot look that would inform us about a larger trial, but our results suggested that wasn’t really needed.  This has been highlighted throughout the manuscript.

It needs to be described about specific rationale (dose od beta-alanine )and references in the section of methods.

Author Response: Thank you for the comment. We have added the rationale for the dose in the supplementation section of the methods.

It also needs of diverse dose groups.

Author Response: Thank you for the comment. We have added information about the total number of participants in each group to the supplementation section of the methods.

Further study ie required utilizing diverse dose (diverse timing) of beta-alanine supplementation on foot ball performance (field test).

Author Response: Thank you for the comment. We have added a sentence to the discussion stating that more information is needed to determine the impact of timed administration of beta-alanine.

Reviewer 2 Report

     The authors investigated the effects of beta-alanine supplementation on body composition, anaerobic performance, and the rating of perceived exertion (RPE) in collegiate rugby athletes. Six-week administration of beta-alanine had no impact on body composition or anaerobic performance, though it did affect RPE during the moments of recovery between intermittent sprints. Many studies have demonstrated both positive and negative effects of beta-alanine administration on physical performance. Though the authors used systematic methods and took original subject-specific approaches in this study, the significance of their findings remains unclear. The physiological significance of their findings seems particularly questionable.

      This study cast doubt on the benefits of beta-alanine supplementation. The authors, however, go on to discuss the potential of beta-alanine in sports performance. The authors should discuss their own results by examining why beta-alanine administration had no impact on the body composition or anaerobic performance of their subjects.

      Fifteen subjects were divided into a beta-alanine-supplementation group and placebo group. The sample of this study was very small, as the authors themselves state in the manuscript. The manuscript also neglects to identify the number of subjects in each group. Readers need to know the number of subjects in each group if they are to understand the results of the study correctly.

      The Discussion is too long and should be drastically revised to describe the significance and hypothesis behind the measurements taken in this study, as well as the points that make this study original.

     Tables 2 and 3 are the same. The authors should show the correct Table 2.

Author Response

Author Responses

Note: All changes to the manuscript are highlighted in yellow.

Reviewer 2

The authors investigated the effects of beta-alanine supplementation on body composition, anaerobic performance, and the rating of perceived exertion (RPE) in collegiate rugby athletes. Six-week administration of beta-alanine had no impact on body composition or anaerobic performance, though it did affect RPE during the moments of recovery between intermittent sprints. Many studies have demonstrated both positive and negative effects of beta-alanine administration on physical performance. Though the authors used systematic methods and took original subject-specific approaches in this study, the significance of their findings remains unclear. The physiological significance of their findings seems particularly questionable.

      This study cast doubt on the benefits of beta-alanine supplementation. The authors, however, go on to discuss the potential of beta-alanine in sports performance. The authors should discuss their own results by examining why beta-alanine administration had no impact on the body composition or anaerobic performance of their subjects.

Author Response: Thank you for your comment. We have added additional information to the discussion regarding our interpretation of the null results.

      Fifteen subjects were divided into a beta-alanine-supplementation group and placebo group. The sample of this study was very small, as the authors themselves state in the manuscript. The manuscript also neglects to identify the number of subjects in each group. Readers need to know the number of subjects in each group if they are to understand the results of the study correctly.

Author Response: Thank you for the comment. We have added the number of subjects in each group to the supplementation section of the manuscript.

      The Discussion is too long and should be drastically revised to describe the significance and hypothesis behind the measurements taken in this study, as well as the points that make this study original.

Author Response: Thank you for the comment. We have reduced the length of the discussion and provided additional information regarding the significance of the findings of this study.

     Tables 2 and 3 are the same. The authors should show the correct Table 2.

Author Response: Thank you for the comment. We have updated Table 2 to show the correct information.

Reviewer 3 Report

This paper reports the results of a double-blinded placebo-controlled study of the impact of beta-alanine supplementation on exercise performance in a sample of college athletes. Strengths of this study include a strong design and the use of a range of relevant outcomes measures. My primary substantive concern relates the sample size and whether it can be demonstrated to be adequate to address the study goals.

The small final sample size makes it difficult to judge whether the lack of significant results is due to the ineffectiveness of the intervention or because the study is under-powered. It is important in this case to include a post-hoc power analysis describing the minimum effect size that could have been detected based on this sample to help understand the capacity of the design to identify meaningful group differences. For example, the difference in the pre/post change in mean recovery blood lactate between groups is 2.5 mmol (13.4 - 10.3 = 3.1 supplementation vs. 12.7 – 12.1 = 0.6 placebo) which appears to amount to a roughly 1SD magnitude of difference – which would probably be clinically meaningful. If the sample size is not sufficient to detect a difference with this magnitude, this is potentially a problem. A power analysis would help to quantify the extent of the issue, and will allow you to explain what conclusions can (and can’t) be drawn from a sample this size. I do not see any report of the specific group sizes of the supplementation and placebo groups. This is also important information to include.

Author Response

Author Responses

Note: All changes to the manuscript are highlighted in yellow.

Reviewer 3

This paper reports the results of a double-blinded placebo-controlled study of the impact of beta-alanine supplementation on exercise performance in a sample of college athletes. Strengths of this study include a strong design and the use of a range of relevant outcomes measures. My primary substantive concern relates the sample size and whether it can be demonstrated to be adequate to address the study goals.

The small final sample size makes it difficult to judge whether the lack of significant results is due to the ineffectiveness of the intervention or because the study is under-powered. It is important in this case to include a post-hoc power analysis describing the minimum effect size that could have been detected based on this sample to help understand the capacity of the design to identify meaningful group differences. For example, the difference in the pre/post change in mean recovery blood lactate between groups is 2.5 mmol (13.4 - 10.3 = 3.1 supplementation vs. 12.7 – 12.1 = 0.6 placebo) which appears to amount to a roughly 1SD magnitude of difference – which would probably be clinically meaningful. If the sample size is not sufficient to detect a difference with this magnitude, this is potentially a problem. A power analysis would help to quantify the extent of the issue, and will allow you to explain what conclusions can (and can’t) be drawn from a sample this size.

Author Responses

Note: All changes to the manuscript are highlighted in yellow.

Author Response: Thank you for your pertinent comments. We acknowledge the potential for this pilot study to be underpowered due to participant attrition. As post-hoc power analyses have been criticized heavily (Hoenig and Heisey 2001; Greenland 2012; Greenland et al. 2016), we have chosen to report confidence intervals and effect sizes for all outcome measures to provide the reader with more information regarding the practical/clinical relevance of our findings.

Greenland S, Senn SJ, Rothman KJ, et al.: Statistical tests, p values, confidence intervals, and power: A guide to misinterpretations. Eur J Epidemiol 2016, 31:337-50.

Greenland S: Nonsignificance plus high power does not imply support for the null over the alternative. Annals of Epidemiology 2012, 22:364-368.

Hoenig JM and Heisey DM: The abuse of power. The American Statistician 2001, 55:19-24.

I do not see any report of the specific group sizes of the supplementation and placebo groups. This is also important information to include.

Author Response: Thank you for your comment. Group sizes have been added to the supplementation section.

Reviewer 4 Report

An investigation of beta-alanine as it pertains to the practical aspects of sports performance.

This is a great premise for the exploration of beta-alanine in practical purposes, but there are some serious flaws in this research.

Training Program– you make this section sound very controlled and formal, however, you have multiple participants who were excluded due to ‘poor exercise technique’.  Was the strength and conditioning coach unqualified? It is also mentioned that participants were allowed to conduct their own exercise regimen with no guidance in the discussion. Why was there no control for exercise? Differences in training may have serious implications for any time effects seen that cannot simply be brushed aside by stating that you do not feel this impacted the results.

Interaction effects can also not be considered for this experiment without final group numbers for placebo and supplemental group.  As it stands, the previous studies the authors use to justify low participant numbers either began with the determinant number to randomize/match or provided the post attrition group sizes.  This combined with the lack of structured exercise during the 6 week supplementation makes the results very unreliable. Due to these factors, more rigor needs to be demonstrated in expressing the true nature of the results.

Writing style:

Consistency with abbreviations – once an item has been introduced and an abbreviation designated use it consistently. Either reintroduce it in each section for clarity or use the abbreviation throughout the paper.

Beta-alanine – introduced as BA in the abstract, and supplementation section, but abbreviation not used through the remainder of the paper same with placebo/PLA

1RM – introduced in both line 94 and 99

USE and LSE introduced in the first section of Methods—and then reintroduced in the sub-section, but abbreviations for BA and PLA not introduced until Supplementation sub-section even though mention in the first section of methods

Specific Corrections:

Title: Not sure why anaerobic exercise is in all caps

Line 9: replace 2 with 3

Line 74: There is a parentheses that does not end

Line 95: replace ‘respectfully’ with ‘respectively’

Author Response

Author Responses

Note: All changes to the manuscript are highlighted in yellow.

Reviewer 4

An investigation of beta-alanine as it pertains to the practical aspects of sports performance.

This is a great premise for the exploration of beta-alanine in practical purposes, but there are some serious flaws in this research.

Training Program– you make this section sound very controlled and formal, however, you have multiple participants who were excluded due to ‘poor exercise technique’.  Was the strength and conditioning coach unqualified? It is also mentioned that participants were allowed to conduct their own exercise regimen with no guidance in the discussion. Why was there no control for exercise? Differences in training may have serious implications for any time effects seen that cannot simply be brushed aside by stating that you do not feel this impacted the results.

Author Response: Thank you for your comments. The participants in this study performed all resistance training under the supervision of a Certified Strength and Conditioning Specialist and all rugby practices under the supervision of the sport coach. Though the athletes were trained by the strength coach and were well-accustomed to resistance exercise, several participants did not exhibit ideal exercise technique during one-repetition maximum tests and were excluded from the study. A breakdown of technique is somewhat expected in athletes who are not accustomed to one-repetition maximum testing, even when trained by a qualified strength coach.

We have clarified line 332 to specify that the athletes were not provided a specific program by the research team. However, all team members performed the same resistance training program and sport-specific practices under the supervision of the same sport coach and strength coach.

Interaction effects can also not be considered for this experiment without final group numbers for placebo and supplemental group.  As it stands, the previous studies the authors use to justify low participant numbers either began with the determinant number to randomize/match or provided the post attrition group sizes.  This combined with the lack of structured exercise during the 6 week supplementation makes the results very unreliable. Due to these factors, more rigor needs to be demonstrated in expressing the true nature of the results.

Author Response: Thank you for your comments. We have added the specific participant numbers in each group to the supplementation section and apologize for the oversight.

Writing style:

Consistency with abbreviations – once an item has been introduced and an abbreviation designated use it consistently. Either reintroduce it in each section for clarity or use the abbreviation throughout the paper.

Beta-alanine – introduced as BA in the abstract, and supplementation section, but abbreviation not used through the remainder of the paper same with placebo/PLA

Author Response:  This has been standardized across the paper.

1RM – introduced in both line 94 and 99

Author Response:  We have removed the second introduction.

USE and LSE introduced in the first section of Methods—and then reintroduced in the sub-section, but abbreviations for BA and PLA not introduced until Supplementation sub-section even though mention in the first section of methods

Author Response: This has been standardized across the paper. 

Specific Corrections:

Title: Not sure why anaerobic exercise is in all caps

                Author Response: Thank you for the comment. We have corrected the title.

Line 9: replace 2 with 3

                Author Response: This has been corrected.

Line 74: There is a parentheses that does not end

                Author Response: Thank you for the comment. This has been corrected.

Line 95: replace ‘respectfully’ with ‘respectively’

Author Response: Thank you for the comment. This has been corrected.

Round 2

Reviewer 2 Report

The authors revised their manuscript in response to my comments.

Author Response

Author Responses

Sports-616851

Reviewer 2

The authors revised their manuscript in response to my comments.

                Author Response: Thank you for your appraisal of our manuscript.

Reviewer 3 Report

My comments on the previous draft have been fully addressed.

Author Response

Author Responses

Sports-616851

Reviewer 3

My comments on the previous draft have been fully addressed.

                Author Response: Thank you for your appraisal of our manuscript.

Reviewer 4 Report

Hello, thank you for your willingness to respond and revise your paper based on previous feedback.

Based upon the author's responses, however, I am no longer able to recommend this paper for publication.

The statement that individuals were removed from the results based upon "poor exercise technique" and the authors' assertions that this was during the testing phase of the experiment and attributed to the inability of athletes trained by a certified professional to perform 1RM testing properly then forces me to call into question the ability of the research team to properly perform and observe the testing procedures.  Previous research has demonstrated the reliability of 1RM testing in both novel lifters and resistance trained individuals (Ritti-Dias et al, 2011, and Levinger et al,2009).  Therefore this speaks either of selective bias by the research team or inadequate ability to properly instruct and monitor the participants during research.  In either case, this severely calls into question the validity of the results from this research.

Additional issues:

If this is a pilot study, the title and abstract need to be changed to reflect this.

If all participants were allowed to maintain protein supplementation during the course of the study a dietary analysis needed to be performed to insure there were no significant difference in protein intake between the groups that might have affected strength and endurance testing.

Author Response

Author Responses

Sports-616851

Reviewer 4

Hello, thank you for your willingness to respond and revise your paper based on previous feedback.

Based upon the author's responses, however, I am no longer able to recommend this paper for publication.

The statement that individuals were removed from the results based upon "poor exercise technique" and the authors' assertions that this was during the testing phase of the experiment and attributed to the inability of athletes trained by a certified professional to perform 1RM testing properly then forces me to call into question the ability of the research team to properly perform and observe the testing procedures.  Previous research has demonstrated the reliability of 1RM testing in both novel lifters and resistance trained individuals (Ritti-Dias et al, 2011, and Levinger et al,2009).  Therefore this speaks either of selective bias by the research team or inadequate ability to properly instruct and monitor the participants during research.  In either case, this severely calls into question the validity of the results from this research.

Author Response: Thank you for your comments. To clarify, the individuals were excluded from the final analysis because of inappropriate exercise technique or effort during post-testing, though they displayed acceptable effort and exercise technique during pre-testing. We apologize for how our feedback could have resulted in you believing that exclusions were occurring due to poor oversight, testing controls, etc. when in fact we made the decision to exclude people from analysis because their effort on the test were deemed to be sub-maximal.  Even though all participants were instructed to perform maximally and had research team members directly supervising the tests, some individual’s post-testing effort was sub-maximal.  Upon interviewing participants after the completion of testing, several participants admitted to not wanting to get injured, fatigued, or sore due to a highly-contested match against a rival team several days after post-testing. These problems were not observed during pre-testing. In this case, the most appropriate choice to ensure the validity of our results was to exclude the non-compliant participants’ data from the final analysis.

Further, to ensure potential readers do not draw the same conclusions that you did, we have decided to move this language about reasons for excluding to the very first part of the results section of our manuscript.  Even though this further reduced our sample size, we feel this decision helped to ensure consistent outcomes with the day were analyzing for this manuscript.  Please see our attached changes.

Additional issues:

If this is a pilot study, the title and abstract need to be changed to reflect this.

Author Response: Thank you for your comments. The title and abstract were changed to emphasize that this investigation is a pilot study.

If all participants were allowed to maintain protein supplementation during the course of the study a dietary analysis needed to be performed to insure there were no significant difference in protein intake between the groups that might have affected strength and endurance testing.

Author Response: Thank you for your pertinent comment. We collected dietary data during the study (initially described on Line 100) but had not provided the results in-text. Our analysis revealed no significant main or interaction effects for absolute and relative kilocalorie, protein, carbohydrate, and fat intake which might have affected the strength and power outcomes collected in this study. That data is now provided.  We are sorry for its omission.